# Biologically-constrained spiking neural network for neuromodulation in locomotor recovery after spinal cord injury

Raymond Chia *, Chin-Teng Lin

Computational Intelligence and Brain Computer Interfaces Lab, School of Computer Science, Faculty of Engineering and Information Technology, University Technology Sydney, Sydney, New South Wales, Australia

* raymond.chia@uts.edu.au

## Abstract

Presynaptic inhibition after spinal cord injury (SCI) has been hypothesised to disproportionately affect flexion reflex loops in locomotor spinal circuitry. Reducing gamma-aminobutyric acid (GABA) inhibitory activity increases the excitation of flexion circuits, restoring muscle activation and stepping ability. Conversely, nociceptive sensitisation and muscular spasticity can emerge from insufficient GABAergic inhibition. To investigate the effects of neuromodulation and proprioceptive sensory afferents in the spinal cord, a biologically constrained spiking neural network (SNN) was developed. The network describes the ankle flexor motoneuron (MN) reflex loop with inputs from ipsilateral Ia- and II-fibres and tonically firing interneurons. The model was tuned to a Baseline level of locomotive activity before simulating an inhibitory-dominant and body-weight supported (BWS) SCI state. Electrical stimulation (ES) and serotonergic agonists were simulated by the excitation of dorsal fibres and reduced conductance in excitatory neurons. ES was applied across all afferent fibres without phase- or muscle-specific protocols. The present computational findings suggest that reducing stance-phase GABAergic inhibition on flexor motoneurons could facilitate more physiological flexor activation during locomotion. The model further predicts that neuromodulatory therapy, together with body-weight support, modulates the balance of synaptic excitation and inhibition in ankle flexor motoneurons to mitigate excessive inhibitory drive in the ankle flexor circuitry.

## Author summary

SCI is a life-altering condition that often leaves young adults paralysed and reliant on others for support. Restoring the ability to walk is a critical goal to help improve independence and quality of life for people living with SCI. Promising new treatments, such as spinal cord stimulation and drug therapies, aim to reawaken the neurons that control walking. However, scientists still do not entirely understand

**Data availability statement:** All relevant data are within the manuscript and its Supporting information files.The experiment code can be found at https://github.com/rchia16/balancing-locomotor-networks.git.

**Funding:** This work was supported by a Research Training Program (RTP) Scholarship provided by the Australian Government and awarded to R.C for the period 2017–2022 to support the completion of his doctoral degree. The funders had no role in study design, data collection and analysis, decision to publish, or preparation of the manuscript. R.C received a stipend salary from the Australian Government under the Research Training Program during this period.

**Competing interests:** The authors have declared that no competing interests exist.

how these treatments work. In this study, we developed a detailed computer model of the neural circuits involved in walking to test how therapies such as serotonin-boosting drugs, ES, and BWS training might help. Our findings suggest that these treatments can work together to reduce excessive inhibition that blocks ankle movement, leading to smoother and more coordinated steps. This research helps uncover how these therapies work and provides insights to develop better rehabilitation strategies for improving walking after SCI.

## Introduction

SCI globally affects an estimated 9 million people as of 2019, with an age standard-ised incident rate of about 109 per 100,000 [1]. In the event of SCI, damage to the nervous tissue can result in loss of voluntary control, sensation, spasticity, diaphragm dysfunction, pressure ulcers, and pain syndromes [2,3]. Sufferers of SCI often report non-physical symptoms such as emotional disorders, loss of independence, depression, anxiety, and clinical levels of stress [4]. The lifelong management places an intense financial burden not only on the patients and their communities, but also on the broader economic landscape [5,6]. Lifetime medical costs in Canada can range from $1.47 to $3 million CAD (2013 prices) per person [7], £1.12 million per person in the UK (2016 prices) [8], and range from $0.77 to $1.3 million USD (1995 prices) in the US [9]. Recovering voluntary muscle activity and returning locomotion activity to SCI sufferers could save societal and patient costs while also improving the patient's quality of life [10,11].

Recovering gait remains a top priority for people living with SCI [12]. Flexor activity is critical for step progression during locomotion, acting as a shock absorber before foot strike [13], adapting step-height to continue locomotion progression [14], and resetting locomotion [15]. Increasing the excitability of locomotor networks after paralysis can improve locomotor capabilities; however, hyperexcitation of flexor muscles can result in spastic muscle expression, leading to poor balance and coordination [16–18]. Maintaining an excitation-inhibition balance of excitability emerges as an intuitive solution to enabling robust locomotor expression.

SCI interrupts normal bidirectional signalling, leading to dysfunctional neural circuitry, tilting the balance of excitation and inhibition [19]. Lack of descending activity keeps MNs at a predominantly inhibited state, while inhibitory populations in the dorsal and intermediate zone become over-reactive [20,21]. A large percentage of the SCI population experience spastic muscle activity, likely due to insufficient release of GABA neurotransmitters [22–26]. Nevertheless, even with an overly excited or inhibited environment and detached from brain inputs, the locomotor spinal circuit can continue to express coordinated motor function given sufficient excitation and contextually relevant sensory information [27–31].

Proprioception is a critical sensory input to entrain and recover locomotion after SCI [32–34]. Proprioceptive afferent innervation is widespread and diverse, project-ing to MNs [35–37], GABAergic [38,39], and serotonergic [40–43] interneurons (INs) in the dorsal and intermediate zone of the spinal cord. Long and short axons spread

across multiple segments and are organised spatially and by modality [37,44–46]. Proprioceptive interneurons (PINs) are mainly excitatory, most inhibitory populations projecting ipsilaterally [47]. Due to their complex and integratory nature, PINs have been suggested to be a possible neural detour around spinal lesions, recovering voluntary sensorimotor control after SCI [46,48–51].

Genetic labelling of spinal cord interneurons has identified V2a interneurons (Chx10$^+$) as a crucial population for coordinating left/right coordination [52] and locomotor speeds [53,54]. V2a interneurons reside in laminae VII and receive serotonergic, glutamatergic inputs from the brainstem and sensory inputs from dI5 INs and mechanosensory feedback [53,55–58]. V2a interneurons locally project bursting glutamatergic excitation to ipsilateral V0 interneurons and MNs [55,59,60]. Moreover, ablation of V2a interneurons has been shown to prevent the recovery of suprathreshold ES facilitated locomotion recovery [61]. Therefore, inclusion of the V2a population in the present model allowed us to investigate how neuromodulation and sensory feedback influence excitatory drive and recovery of flexor motor function after injury.

After SCI, axons spared from injury allow voluntary activation and sensation of the body past lesion sites [51,62]. Traditional rehabilitation therapy leverages these residual connections to maximise motor skills via therapeutic exercise or overcome losses with assistive devices [28,63]. Neuromodulation techniques such as spinal cord ES [30,31,64,65] and pharmacology [51,66,67] have shown to recover locomotor activity after SCI. Moreover, chronic application of ES in conjunction with physical rehabilitation enabled volitional muscle activation even without ES [29]. Although these observations show promise for new and effective neurorehabilitation therapies, the mechanisms of action and synergy between sensory ensembles and ES remain in question [68,69].

It is natural to seek methods to return excitation to sub-lesional networks after losing descending input [70]. Most ES techniques have sought to excite and entrain locomotion by activating dorsal roots in the epidural space [69,71–74]. However, ES, in the same anatomical space, can also evoke inhibition and restore balance to an overly excited network [23,24,75]. A more in-depth and nuanced view of ES therapy is required to fully appreciate the complexity of modulating the neural environments. A key question remains: How do neuromodulation therapies integrate with sensory information? We hypothesise that spinal cord locomotor circuits require balanced excitation and inhibition to coordinate flexor activity.

Understanding excitation and inhibition balance experimentally remains challenging due to the complexity of interacting neural pathways. Computational modelling offers a complementary method to test how specific neural architectures and synaptic mechanisms contribute to locomotor activity.

Computational models have been central in linking neural architecture to locomotor output. Studies of commissural and long propriospinal interneurons have helped elucidate how propriospinal circuits stabilise gait across speeds [76]. Building on these frameworks, combining modelling and experimental approaches has revealed the role of V3 neurons in speed-dependent left-right interlimb coordination [77].

These models feature detailed architectures but often rely on reciprocally inhibiting phase oscillators [78,79] driven by tonic inputs that mimic descending mesencephalic locomotor region activity [80–82]. While suitable for neurologically intact systems, such assumptions break down after SCI, where the supraspinal drive is reduced or absent, although sub-lesional afferents remain intact.

To address this limitation, several studies have integrated peripheral feedback into spinal locomotor circuits, yielding physiologically grounded spiking models and enabling the design of novel neuromodulation therapies [69,71,74,83]. A bio-inspired controller combining a balance controller, a central pattern generator (CPG), and a sensory feedback network reproduced human gait kinematics and ground reaction forces by optimising for effort and stability [83]. Similarly, by combining spiking networks with finite element modelling, scientists were able to experimentally converge model outputs to rodent ES results, revealing activation sites and recruitment dynamics [71]. Incorporating locomotion-specific afferents into these models further enabled the development of function-specific ES therapies [69] and clarified interactions between suprathreshold ES and proprioceptive afferents [74].

Iterative refinements of CPG-based frameworks have expanded our understanding of locomotor control and informed new therapeutic strategies. However, the current spinal locomotor network modelling landscape omits presynaptic inhibition.

This study aims to understand the effect of SCI-induced imbalanced presynaptic inhibition in sensory-driven rodent locomotor spinal networks. We describe a biologically constrained tibialis anterior (TA) SNN rodent model receiving heterogeneous excitatory and inhibitory synapses, including GABAergic presynaptic inhibition. A combination of ES and serotonin agonist (5-HT) neuromodulators are simulated in an SCI and SCI with BWS locomotion setting. We show that combining BWS with ES reduces overactive stance-evoked GABA inhibition and returns TA MN firing rates towards Baseline.

## Methods

A biologically constrained SNN was developed to investigate neuromodulation effects on sensory-driven rodent spinal locomotor circuits. Simulations were run on an Intel Xeon Gold 6238R 2.2GHz Processor. The software was developed in Python 3.10.0 using the Brian2 neural simulator module (v2.6.0) [84]. The simulation time step was set to $50\,\mu s$ and Euler approximations for ordinary differential equation solving. A total of eight locomotor steps were simulated, where gait stance and swing phases were split at 65% of the gait cycle [85]. This study simulated three different neurological environments, including a Baseline, SCI, and SCI with a BWS state. Each neurological state was modulated with inputs from ES and 5-HT. The Baseline state was set by validating the static outputs of the SNN model against previously validated computational data and experimental observations in healthy rodents. SCI condition was set by increasing the paired synapses between GABA INs and TA MNs by a factor of 1.6, as experimentally reported in previous rodent SCI studies [86]. Finally, the BWS state was defined as the scalar reduction of Ia and II afferent firing rates as reported from treadmill BWS experiments recording EMG [87].

The SNN model architecture was biologically constrained using synaptic connections inferred from previous cell staining [39,71,86,88], electrophysiological [53,89–94], and genetic works [55,95–99]. A second biological constraint was set by matching neuron cell dynamics to experimentally measured electrophysiological neural recordings [53,55,74,88,95, 97,100–102]. Specific parameter settings are described in the sections below and set to be within biologically plausible ranges. Parameter definitions have been summarised in Table 1. Finally, propriosensory inputs were constrained to previously validated musculoskeletal and muscle spindle models [74].

Simulated data were first tested for normality using the Shapiro-Wilk test, where $p > 0.05$ indicated that the data did not significantly deviate from a normal distribution. Normally distributed data were compared using a paired t-test, while non-normal data were analysed using the Wilcoxon signed-rank test. Distributions were considered significantly different if $p < 0.05$

Equivalence between groups was assessed using pairwise two one-sided tests (TOST) at an $\alpha_{equiv} = 0.05$ level (95% CI). The null hypothesis of non-equivalence was rejected if both one-sided tests were significant ($p_{equiv} < 0.05$), indicating that the mean difference lay within the predefined equivalence region ($\pm 15\,Hz$). The region of practical equivalence (ROPE) for comparisons between conditions was defined as half the minimal detected burst rate for TA motoneurons in rodents at rest ($30 - 60\,Hz$) [103]. Within-timestep differences were tested for significance using within-step permutation testing with 2000 repetitions.

The experiment code can be found at https://github.com/rchia16/balancing-locomotor-networks.git.

## Afferent signal inputs

Ia and II TA and gastrocnemius medialis (GM) muscle afferent signals were calculated by using musculoskeletal and muscle spindle models during locomotion. The signals were retrieved from the publicly accessible GitHub repository associated with the original publication [74]. To emulate BWS afferent signals, both TA and GM Ia and II data were offset by a

**Table 1**. **List of definitions relevant to the description of the SNN model.**

| Parameter | Definition |
|---|---|
| N | Number of axons/neurons |
| $\tau_{mem}$ | Membrane time constant |
| $\tau_{ref}$ | Refractory period |
| $\tau_{\gamma}$ | GABA spillover time constant |
| V | Membrane potential |
| $E_l$ | Reverse potential |
| $V_{th}$ | Threshold potential |
| $V_{reset}$ | Membrane potential after spike |
| $C_m$ | Membrane capacitance |
| $r_{axon}$ | Axon radius |
| l | Length |
| $g_L$ | Leak conductance |
| I | Input current from noise or synapses |
| w | Adaptation variable |
| $\tau_w$ | Adaptation variable time constant |
| $\Delta_V$ | Sharpness of action potential initiation |
| a | Voltage coupling parameter |
| b | Spike triggered adaptation value |
| $p_{syn}$ | Synaptic connection probability |

scalar amount using values from electrophysiological experiments [87]. The new BWS equations were set to Eq (1) where $K_{GM} = -0.6$ and $K_{TA} = -0.122$, scaling the EMG envelope to 40% and 87.8% respectively, simulating 60% effective body weight. This corresponds to an effective reduction in Ia and II afferent firing rates of approximately 60% of GM muscle afferents and 12.2% for TA muscle afferents, consistent with experimentally observed muscle-specific reductions in EMG amplitude under 60% BWS [87].

EMG was adapted from past computational studies and calculated as per S2 Algorithm [74]. The equations refer to $x_{stretch}$ as stretch, $v_{stretch}$ as stretch velocity, and $EMG_{env}$ as the min-max normalised EMG envelope [104]. Since $EMG_{env}$ magnitude ranged between 0 and 1, a scalar offset can be applied to the afferent signal inputs [74]. Afferent signals were set as timed array Poisson-distributed inputs with a sampling frequency of 200 Hz, and connected to leaky integrate and fire (LIF) axon models, see Eq 3 and Fig 1A and 1B. Parameters were set according to previously validated computational models [71] and tuned to replicate the input firing rate, see Table 2 and Table 5. Background noise of afferent axons ($I_{noise}$) was modelled as a normally distributed variable with a standard deviation scaled to 0.3 pA. Tuning was validated with Pearson correlation coefficient and the mean absolute error. For detailed results, refer to S1 Table in the Supplementary Information. For a complete algorithmic description of a LIF model, see S1 Algorithm.

$$\text{Ia firing rate} = 50 + 2x_{stretch} + 4.3 \cdot sign(v_{stretch}) \cdot |v_{stretch}|^{0.6} + K \cdot 50 \cdot EMG_{env} \tag{1}$$

$$\text{II firing rate} = 80 + 13.5x_{stretch} + K \cdot 20 \cdot EMG_{env} \tag{2}$$

$$\frac{dV}{dt} = \frac{E_l - V}{\tau} + \frac{I_{noise}}{C_m \cdot \pi r_a l} \tag{3}$$

## Spiking neural network

The SNN model simulated the ipsilateral rodent ankle flexor's mono- and di-synaptic stretch and stretch velocity afferent reflexes. Proprioceptive afferents innervated the TA MN, GABA, Ia inhibitory, and V2a interneurons [107,108]. Ia inhibitory interneurons (IaINs) receiving Ia and II afferent inputs of the flexor and extensor muscles were reciprocally inhibited [96]. GABA INs applied presynaptic inhibition to excitatory inputs of the TA MN [86,109]. V2a INs received flexor II

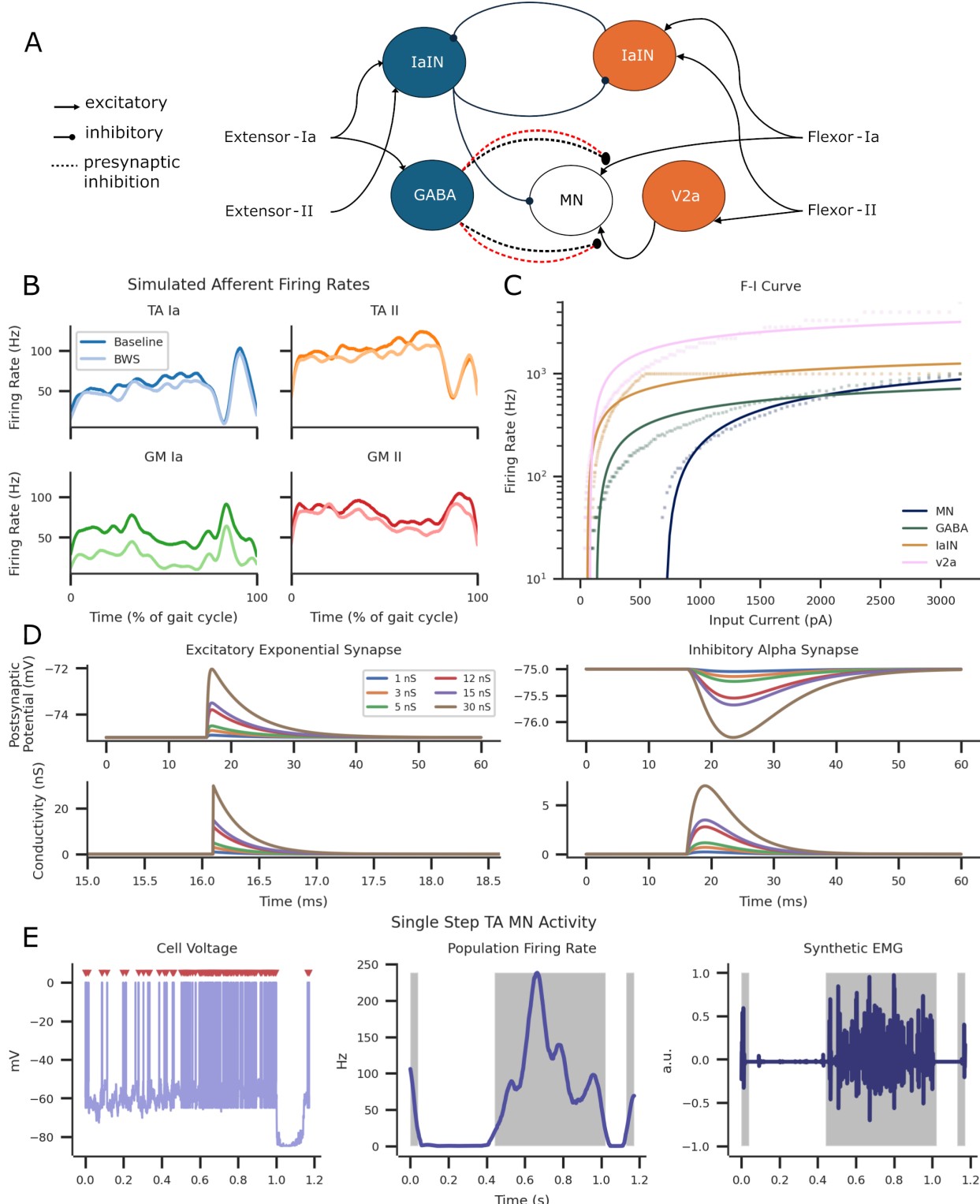

**Fig 1. Computational SNN model of the flexor network with GM extensor and TA flexor proprioceptive Ia and II inputs.** (A) Biologically constrained SNN ankle flexor model. Arrow ends indicate excitation, circle ends indicate inhibition, and dotted line with circle ends indicate presynaptic inhibition connections. The red dotted lines represent an increased number of SCI-induced GABA synapses. (B) Afferent axon firing rates for extensor (GM) and flexor (TA) Ia- and II-fibres in the Baseline and BWS condition. (C) Frequency-Current (F-I) curve for each neuron simulated in the SNN

model. Each point reflects a single data point for the injected current and the solid lines were fitted using a logarithmic function. Input current was applied with a 200 ms pulse width from a stimulus range of 10 to 3160 pA. (D) Excitatory exponential synapses and inhibitory alpha synapses across different conductances. (E) Simulated TA MN output from a single step. Left illustrates the MN membrane potential activity with spiking activity indicated with red triangles. The firing rates were converted into 25 ms window widths and smoothened with a Gaussian filter (middle). Synthetic EMG activity was generated from recorded spiking activity by convolving Gaussian wavelets (right). Grey shading highlights detected bursts of firing rate activity.

Table 2. **Afferent axon parameters for LIF model, including references for each value where relevant.**

| Axon Parameter | Value | Reference |
|---|---|---|
| N | 60 | [69] |
| $\tau$ | 30 ms | [69] |
| $\tau_{ref}$ | 1.6 ms | [69] |
| $E_l$ | −80 mV | [71] |
| $V_{th}$ | −60 mV | [105] |
| $V_{reset}$ | −70 mV | [105] |
| $C_m$ | 1.0 µF/cm$^2$ | [106] |
| $r_{Ia}$ | 9 ±0.2 µm | [71] |
| $r_{II}$ | 4.4 ±0.5 µm | [71] |
| $l^\dagger$ | 1 mm | - |

$^\dagger$ Assumed variable, held constant throughout tuning.

afferent inputs and applied tonic excitation to TA MNs [55,59]. TA MNs received monosynaptic excitation from flexor Ia afferents [98,110]. Refer to Fig 1A for an illustration of the entire network and Fig 1C for frequency-current (F-I) response using 200 ms pulse widths across a stimulus range of 10 to 3160 pA. For a visual example of tonic spiking neural firing, refer to S2 and S3 Figs.

IaINs were modelled as conductance-based LIF neurons receiving excitation from Ia and II afferent fibres and inhibition from opposing IaINs, see Eq 4. $I_{syn}$ is the cumulative synaptic current from excitatory and inhibitory components. IaIN parameters in Table 3 were set to match experimental results [74,88].

$$\frac{dV}{dt} = \frac{g_L(E_L - V) + I_{syn}}{C_m}$$

(4)

Table 3. **Ia inhibitory, GABA, V2a IN and MN parameters.** Parameter values were set based on results recorded in experiments or other validated computational studies.

| Parameter | IaIN | GABA | V2a | MN |
|---|---|---|---|---|
| N | 196 [74] | 196$^\dagger$ | 196 [74] | 169 [69] |
| $C_m$ | 31.1 pF [88] | 100 pF$^\ddagger$ | 45 pF [55] | 162 pF [111] |
| $E_L$ | −70 mV [88] | −70 mV$^\dagger$ | −53 mV [55] | −75 mV [71,97] |
| $V_{th}$ | −50 mV$^\dagger$ | −50 mV$^\dagger$ | −42 mV [55] | −50 mV [111] |
| $V_{reset}$ | −65 mV [88] | −62.3 mV$^\dagger$ | −47 mV [55] | −65 mV$^\ddagger$ |
| $g_L$ | 5 nS [88] | 1.2 nS$^\ddagger$ | 1.2 nS [55] | 27 nS [112] |
| $\Delta_v$ | − | 2 mV$^\ddagger$ | 0.5 mV$^\ddagger$ | 0.05 mV$^\ddagger$ |
| a | − | 2 nS$^\ddagger$ | 2 nS$^\ddagger$ | − |
| $\tau_w$ | − | 20 ms [97] | 55 ms [55] | − |

$^\dagger$Assumed variables, held constant during tuning
$^\ddagger$Tuned parameters, adjusted during model calibration.

GABA presynaptic inhibitory INs and V2a INs were modelled as conductance-based adaptive exponential (AdEx) LIFs [101]. GABA parameters [97] and V2a parameters [53,55,95] were set as per experimental results. AdEx equations were defined as per Eqs (5) and (6). GABA and V2a IN parameters were set per Table 3.

$$\frac{dV}{dt} = \frac{g_L(E_L - V) + g_L(\Delta_V \exp \frac{V - V_{th}}{\Delta_V}) + I_{syn} - w}{C_m} \tag{5}$$

$$\frac{dw}{dt} = \frac{a(V - E_L) - w}{\tau_w} \tag{6}$$

TA flexor MNs were modelled as exponential LIFs receiving excitation from TA Ia fibres, V2a INs, inhibition from GM originating IaINs and presynaptic inhibition from GABA INs [38,58,74]. The equation for TA MNs was the same as Eqs (5) and (6) where $w = 0$. Parameters in Table 3 were set to best estimate experimental results [88,111,112]. MN responses under varying stimulation pulse widths are shown in S2 Fig.

To capture the net excitability increase produced by 5-HT agonists, we used a single-parameter surrogate at the motoneuron: a reduction of the leak conductance ($g_L$). This mirrors prior network models that emulate 5-HT with a single conductance change (e.g. 40% reduction of K(Ca)) to reproduce persistent inward current mediated plateaus and enhanced recruitment [69,113]. Consistent with these precedents, MN and V2a IN $g_L$ was reduced by 40% under SCI$_{5-HT}$ and SCI$_{5-HT+ES}$ conditions and by 15% under BWS$_{5-HT}$ and BWS$_{5-HT+ES}$, reflecting the modulated serotonergic activity after 5-HT administration. Decreasing the leak conductance increases membrane resistance and neuronal excitability, thus simulating the modulatory effects of 5-HT on spinal circuits.

## Synapses

Alpha and exponential conductance synapses were used to describe inhibitory and excitatory synapses, respectively, see Table 4 and Fig 1D. The reversal potential of excitatory synapses was set to 0 mV, while inhibitory synapses were set to $-75$ mV [69]. This was to ensure the hyperpolarisation without instability. II-fibre synapse weights were scaled by a factor of 0.33 to simulate the effect of smaller axon size [74,114].

Synaptic connections, with the exclusion of GABA, were determined by probabilities specified in Table 5. Axon synapses included a $2\pm0.3$ ms delay accounting for diameter variability [115]. GABA connections to TA MNs were predetermined by index rules dependent on the experiment condition. Synaptic connections between GABA and TA MN indexes were joined together if their absolute index difference was less than 4, see S1 Fig. GABA connections were tuned to qualitatively match the population firing profile of previously reported simulations using the same dataset [69,74]. SCI condition GABA connections were increased by 160% as seen in flexor MNs after SCI transection, simulating an inhibitory-dominant environment [86].

Presynaptic inhibition was a multiplicative gain, scaling synaptic weights of each excitatory connection to the TA MN population [91,97,116,117]. GABA spillover was modelled as a linear decrease in release factor, $\gamma$, see Eq (7). $\lambda$ determined the strength of the inhibition and set to a value of 0.4. $C_{GABA}$ was considered a unitless value for local GABA

**Table 4. Alpha and exponential synapse and GABA spillover parameters.** Alpha synapses were used for inhibitory connections while exponential synapses were used for excitatory connections.

| Synapse Parameter | Value | Reference |
|---|---|---|
| $\tau_{exc}$ | 0.25 ms | [74] |
| $\tau_{inh,rise}$ | 2 ms | [74] |
| $\tau_{inh,decay}$ | 4.5 ms | [74] |
| $\tau_\gamma$ | 20 ms | [97] |

**Table 5. Synapse connection probabilities ($p_{syn}$) and synaptic conductance by source and target neurons.** Excitatory (exc.) and inhibitory (inh.) synaptic conductance applies to target neurons. All values in this table were tuned to reproduce afferent signals and keep stability during afferent driven locomotion.

| Source | Target | $p_{syn}$ | Exc. | Inh. |
|---|---|---|---|---|
| Axon | IaIN | 0.3 | 7 nS | – |
| Axon | GABA | 0.4 | 12 nS | – |
| Axon | V2a | 0.6 | 1 nS | – |
| IaIN$_{TA/GM}$ | IaIN$_{TA/GM}$ | 0.1 | – | 3 nS |
| Axon, V2a, IaIN | MN | 0.3 | 30 nS | 10 nS |

concentration [118].

$$\tau_\gamma \frac{d\gamma}{dt} = -\gamma + \max(0, \min(1 - \lambda \cdot C_{GABA}, 1)) \qquad (7)$$

Poisson distributed stimulus inputs were used to simulate subthreshold ES at a frequency of 40 Hz, amplitude of 10 mV, and pulse width of 0.2 ms, consistent with experimental protocols for locomotor facilitation [119–121]. This assumption is supported by previous studies demonstrating that exogenous excitatory drives can be represented as asynchronous Poisson inputs in cortical structures [122,123]. Each stimulation input was connected to three non-overlapping afferent fibres.

Photostimulation paradigms provide an experimentally controlled and reproducible framework for precise activation of neuronal populations under static and well-defined stimulation conditions [124,125]. Recent work has demonstrated that recurrent networks fitted to electrophysiological data can fail to predict responses to optogenetic perturbations, indicating incorrect underlying circuit mechanisms, whereas biologically grounded SNNs generalise more reliably to unseen optogenetic perturbations [126]. This establishes photostimulation as a principled means for validating the causal mechanisms implemented in SNN models.

To our knowledge, no validated SNN models have explicitly included presynaptic GABAergic inhibition in a controlled manner. Given that photostimulation responses provide an appropriate benchmark for assessing simulated effects. Therefore, we evaluated GABA interaction by testing if the model reproduced physiologically consistent firing patterns. This was done across random initialisations and beyond non-perturbed conditions.

Validation of GABAergic interactions was performed by adopting electrophysiological evoked-response experiments from intracellular recordings of the TA MN [97]. To simulate photostimulation of GABA neurons, a threshold current pulse was injected using the same pulse durations (1 ms), frequencies (50 Hz), and delays (45 ms) reported in the experimental protocol [97].

To validate the SNN model, an EMG signal (Fig 1E) was generated by convolving representative motor unit action potentials ( S2 Algorithm), using the same parameters as experimental and previously validated computational models [69,74,127]. Population firing rates were averaged with a 25 ms Gaussian window for smoothening.

## Bayesian modelling for seed equivalence

Given the random nature of the synaptic connections and background noise, the eight-step simulations were repeated across 32 random seeds. Repeating simulations across random seeds was performed to ensure the parameters were not overfit to a singular random seed and to improve the robustness of the study. To test for overfitting in the Baseline condition, equivalence testing across all seeds was performed with Bayesian inference modelling. Conditional and seed-dependent testing was completed using Bayesian hierarchical Linear Mixed Models (LMMs).

Temporal resolution was reduced by calculating mean firing rates for each step, and time (x-axis) was rescaled to [0,1] and expanded with a cubic B-spline bases $\beta(t_i)$. A spline for time and seed-specific random effects, nested within conditions, was fitted as per Eq (8). Let $y_i$ be the output for observation $i$ from seed $s(i)$ in condition $c(i)$ at step $t_i$.

$$y_i = a_{c(i)} + a_{s(i)} + B(t_i)^\top \beta_{c(i)} + B(t_i)^\top \beta_{s(i)} + \epsilon_i$$
$$\epsilon_i \sim \mathcal{N}(0, 5)$$

(8)

In this equation, $a_c$ represents the condition-level intercept and $B^\top \beta_c$ is the shared condition-level temporal trajectory. Seed effects were nested within condition and written in a non-centred parameterisation. We used weakly informed information priors $a_c \sim \mathcal{N}(0, 10)$, $\beta_c \sim \mathcal{N}(0, 2I)$, $\sigma_{LMM} \sim HalfNormal(2)$, and $\tau_{a,c}, \tau_{\beta,c} \sim HalfNormal(1)$. Posterior inference was performed with No-U-Turn Sampler (NUTS) method in PyMC [128,129]. Spline degree-of-freedom was set to 3, and the number of spline knots was set to 4. Regression was computed to minimise the root mean square between observations and model outputs. LMMs were considered well fit if no divergences were detected, all Gelman-Rubin statistics (chain convergence) were within the threshold ($\hat{R} < 1.05$), and posterior predictive fit reached threshold ($R^2 > 0.7$) [130].

$$a_s = z_{a,s}\tau_{a,c(s)}, \quad z_{a,s} \sim \mathcal{N}(0, 1)$$
$$\beta_s = z_{\beta,s}\tau_{\beta,c(s)}, \quad z_{\beta,s} \sim \mathcal{N}(0, I_p)$$

(9)

We considered the SNN model to be equivalent across seeds if, at an intercept level $P(\tau_{a,c} < \epsilon_{int}) \geq 0.9$, and at a dynamics level, $P(\tau_{\beta,c} < \epsilon_{dyn}) \geq 0.9$. The ROPE for both LMM intercepts was defined as $\epsilon_{int} = 5\,Hz$. This was calculated as 16.67% of the minimum recorded TA firing rate ($30\,Hz$) during quiet resting activity [131] and standard deviations seen in locomotor activity experimentally recorded in rodents [132]. The ROPE for dynamic trajectory was defined as $\epsilon_{dyn} = \max(5, 0.05\max(y_c(t)))$, scaling with the fitted maximum rate amplitude. A $5\,Hz$ floor was included to avoid vanishing thresholds in low-amplitude conditions, matching the relative temporal dynamics recorded during locomotion [131,133]. By setting a low TA firing rate during quiet resting activity in healthy rodents, we can more confidently assess seed
equivalence across conditions.

To avoid artificially inflating the effective sample size in subsequent statistical analyses, condition-dependent data were aggregated by averaging across all seeds within each step for a given condition. For inter-step analyses, each step's state was inherently dependent on the preceding states and their corresponding seed configurations; therefore, inter-step data were computed independently within each seed to preserve their dependency structure. Outliers were removed by using a z-score threshold of 3 standard deviations from the mean.

## Results

Bayesian hierarchical statistical modelling was applied to fit the nested data structure for equivalence analysis across 32 random seed initialisations and eight steps. To ensure reliable inference, model fit was first assessed by comparing the observed and model-predicted firing rate distributions. Posterior predictive checks showed strong agreement between observed and predicted values, indicating a satisfactory model fit. Although the model underestimated the exponential behaviour near zero firing rates, see Fig 2B.

Model convergence and sampling stability were confirmed through multiple diagnostics. The posterior sampling energy diagnostic density plot demonstrated close agreement between the transition and marginal energy densities, indicating well-mixed chains (Fig 2B). Convergence diagnostics further supported stable sampling, with Gelman-Rubin statistics ($\hat{R} \leq 1.01$) and Bayesian fraction of missing information (BFMI >0.9) within acceptable ranges across all chains, see S2 Table.

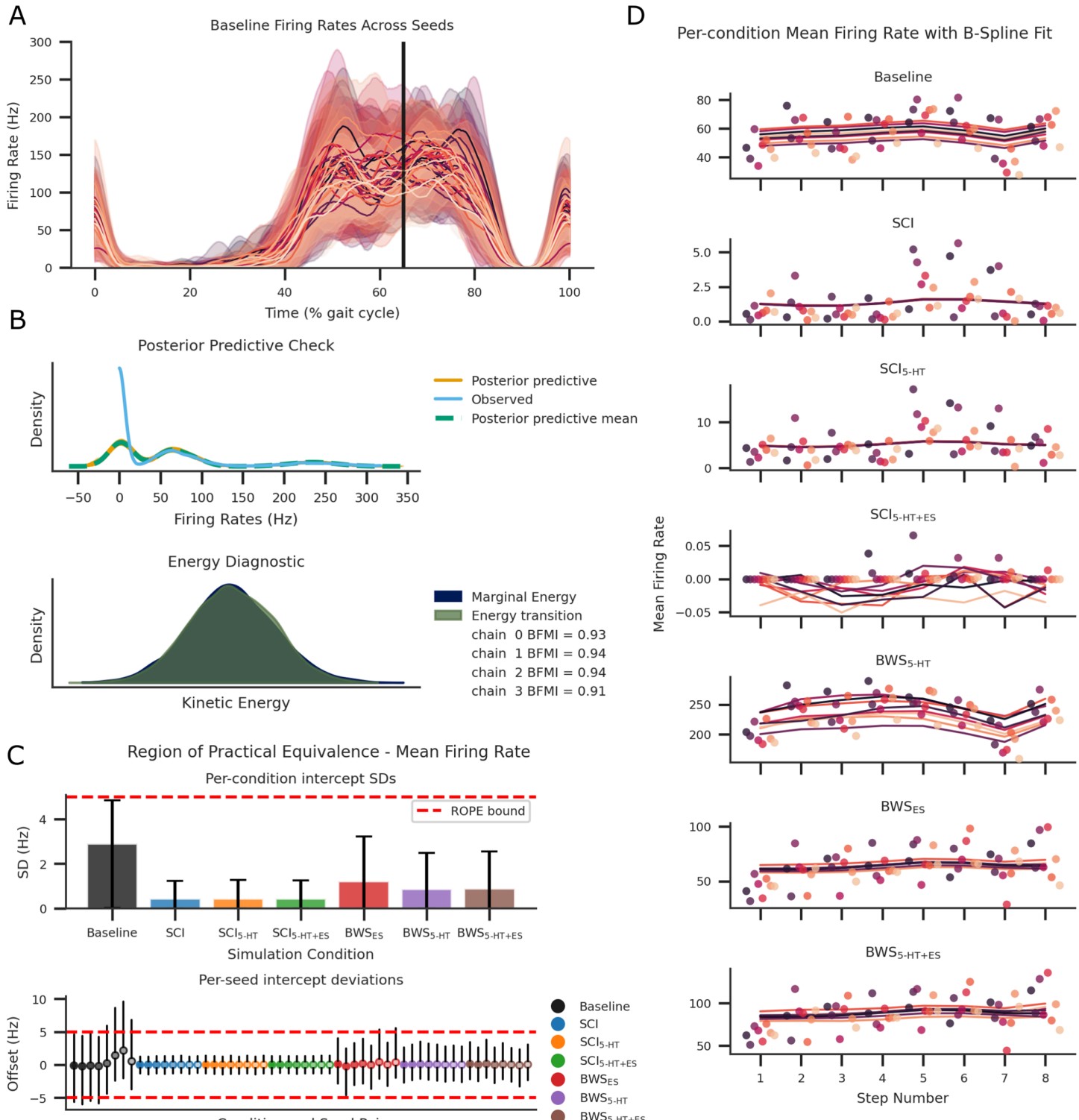

**Fig 2**. **Step trials ($n_{steps}$ = 8) were repeatedly simulated using 32 unique random seed value initialisations.** Recorded TA MN cell spiking events were used to verify model equivalence across all random seeds. (A) The mean population firing rate and standard deviation of TA MNs across the step trials were calculated for each seed, represented by each colour. (B) The upper plot shows the posterior predictive check. Overlap between observed and model-predicted firing-rate densities indicates good model fit. The lower plot illustrates the energy diagnostic of Markov Chain Monte Carlo sampling. The close overlap between marginal and transition energies indicates stable sampling behaviour for the fitted LMM. (C) The posterior

standard deviation (SD) of seed-level intercept deviations within each condition are shown in the bar plot. These are further differentiated within the lower scatter plot as seed-specific intercept deviations. (D) Individual mean firing rates per step are plotted as individual markers. The fitted B-Spline LMM for each seed are colour matched line plots. Sub-figures (C) and (D) only plot eight seeds per condition to avoid overpopulating the figure.

Together, these results confirm that the hierarchical LMM converged robustly and provides a reliable basis for statistical inference. Seed-level intercepts and seed-specific deviations were contained within the ROPE, see Fig 2C. Additional results on seed equivalence across steps and conditions are presented in S3 Table. In summary, aggregating temporal firing rates to mean values per seed and step produced well-converged models that demonstrate equivalence across conditions.

To complement this statistical validation, a spinal reflex recruitment protocol was implemented to verify the physiological plausibility of interactions between neural populations, synapses, and stimulation inputs within the SNN framework. Early- (ER), middle- (MR), and late-response (LR) latencies were defined as 1 ms, 4 ms, and 7 ms respectively [133]. The protocol was adapted from previous animal and simulation experiments and qualitatively assessed model behaviour under static conditions [69,134]. Afferent fibre and direct motoneuron recruitment data, retrieved from validated computational models [74], were coupled with identical stimulation profiles [71]. Electrical pulses of 0.5 ms were applied with an interstimulus interval of 1 s, with stimulus amplitudes ranging from 20 to 600 µA, encompassing the parameters $(100 - 300 \, µA)$ shown experimentally to facilitate locomotion in rats [66,134].

The SNN model described in the methods section does not couple spatial distances or electrode geometry. To match the acquired recruitment data, the minimal required stimulus required to elicit a spike was determined for each neuron and applied to the binary matrix encoding neuron index over time. As shown in Fig 3A, increasing stimulation intensities progressively recruited efferent fibres and suppressed MR and LR amplitudes, consistent with previously reported simulations and in vivo recordings [69,74,134]. Although simulated EMG amplitudes lack physiological scaling, the relative shape, latency, and modulation of evoked responses closely matched experimental [134] and validated computational data [69,71], supporting the model's biological realism.

Additionally, GABA interaction validation was performed by adopting electrophysiological evoked response experiments in EPSP recordings of the TA MN (Fig 3B). The ratios of EPSPs during control and photostimulation conditions were within range of experimental results $(0.5 < EPSP_\lambda / EPSP < 0.8)$ [97].

The SNN model was further verified within a dynamic setting (Fig 3C) using previously simulated afferent fibre firing rate profiles during locomotion [74]. MN firing rates were processed with the same window size (4 ms) used in rodent experiments and validated by comparing TA MN burst firing rates and periods during locomotion [103,135]. The simulated burst activity during locomotion fell within the experimentally reported ranges for burst firing rates $(100 - 500 \, Hz)$ [103] and burst period $(0.5 - 0.9 \, s)$ [135], supporting the model's ability to reproduce physiologically realistic locomotor dynamics.

Having established the model's validity under dynamic conditions, we next examined how network activity evolved across Baseline and SCI simulations averaged over multiple steps. During Baseline stepping, most variation occurred during the swing phase and at the transition between the swing and stance phases of the gait cycle, see Fig 4A. The TA MN population firing rates between Baseline and SCI conditions were significantly different across time (Fig 5A). Deterministically scaling the number of GABAergic synapses onto flexor MNs by 160% increased GABA scaling factor, $\gamma$, which reduced seed-aggregated mean firing rates by a factor of 50 and increased the coefficient of variation by a factor of 3.8, see Fig 4B and Table 6.

Simulated TA MN expression during the Baseline step cycles was attenuated with stimulation voltages greater than 20 mV and by stimulation frequencies greater than 60 Hz, refer to Fig 4C and 4D. Though, GABA and V2a INs firing rates were scaled according to stimulation frequency. While seed variation was less noticeable in interneurons, the TA MN saw a large variation during frequency sweeps greater than 60 Hz during stance to swing transition and swing periods.

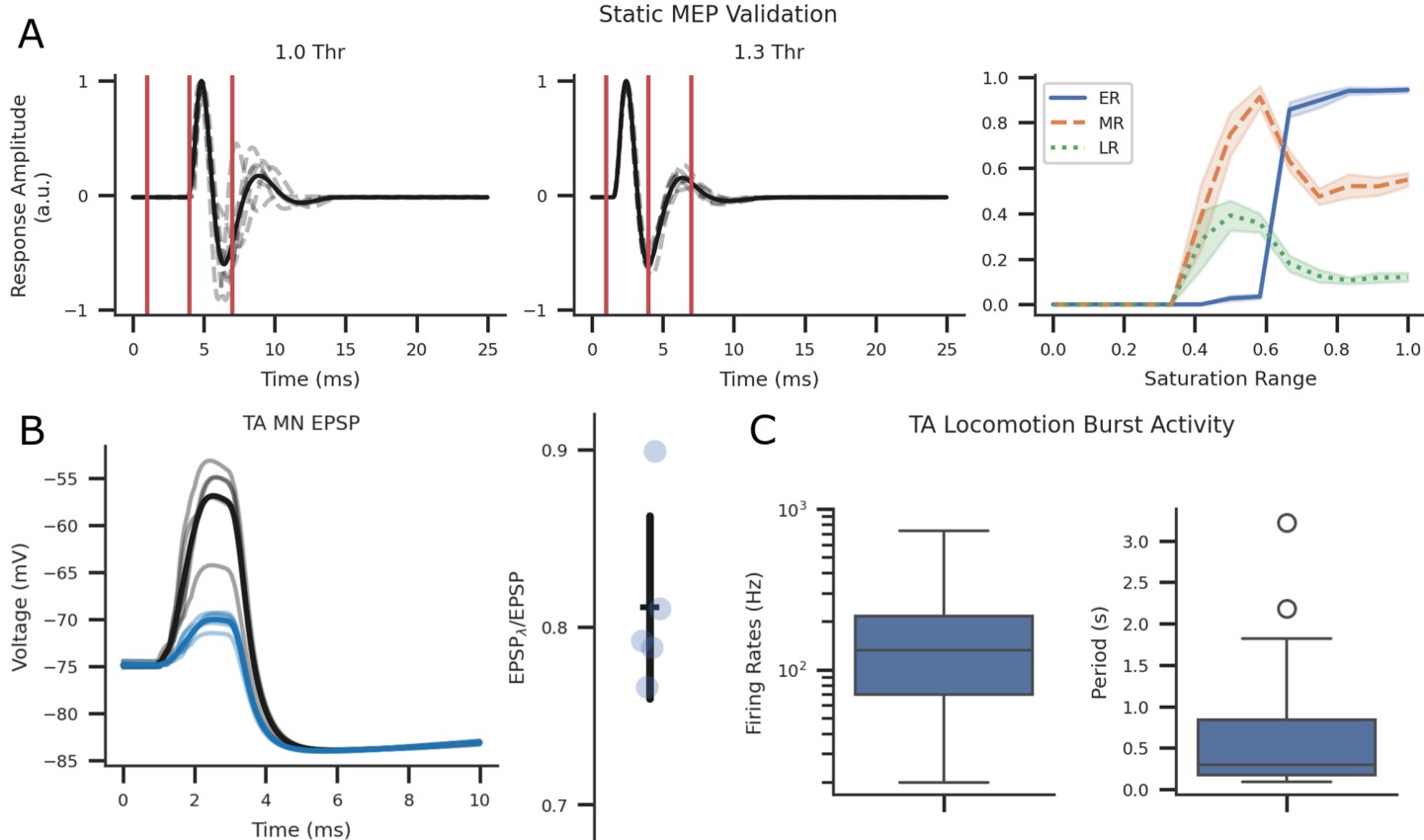

**Fig 3**. **Static (n = 7) and dynamic (n = 9) qualitative and quantitative validation of the SNN model.** (A) Motor evoked potentials (MEPs) at stimulation intensities ranging from 0-600 µA and 0.5 ms pulse width with 1 s between each pulse [69,134]. Results qualitatively match reported findings. (B) Modulation of excitatory post-synaptic potential in control (black trace) and GABA IN stimulation (blue trace) experiments. Transparent traces display seed specific trials, and opaque lines indicate the average across seeds. Fifteen 1 ms pulses at 50 Hz with a 45 ms delay stimulated GABA INs before delivering afferent fibre stimulation at 1.1x threshold [97]. The ratios between the photostimulated and control peaks were within the range of experimental results (0.4-0.9). (C) TA burst firing rates and periods during Baseline locomotion were within the range of experimental results [103,135].

Statistical analysis on flexor activity after outlier removal during each phase presented a non-normal distribution in the swing and stance phases (p < 0.05). Swing phase flexor activity significantly differed between all simulated conditions, with the exclusion of Baseline-BWS$_{ES}$ only. Stance phase flexor activity was significantly different in all simulated conditions. See Fig 5A and 5D for TA MN activity distributions in both swing and stance phases. Practical equivalence testing using an equivalence boundary of ±15 Hz, performed with TOST procedure on firing rate data, revealed equivalence between Baseline-BWS$_{ES}$ pairs during stance phase (see Fig 5D and S4 Table).

Comparison of firing rate differences between the stance and swing phases, where stance was positive and swing was negative, showed significant differences in all Baseline pairs except for Baseline and BWS$_{5-HT+ES}$ condition (see Fig 5A and 5C). The same pattern was reflected in TOST equivalence tests (see Fig 5D and S4 Table.).

Pearson's correlation coefficient was calculated after aggregating the firing rate by averaging across seeds for each condition, see S4 Fig. Pairwise comparison of the Baseline condition with all other simulations found the highest correlation with BWS$_{ES}$, although only weakly correlated ($r = 0.11$; p <0.0001).

While the GABA IN firing rates were the same between the simulated Baseline and SCI settings, increased GABAergic connections resulted in more frequent presynaptic inhibition activity on TA MN populations, see Figs 4B, 5A, and 5B.

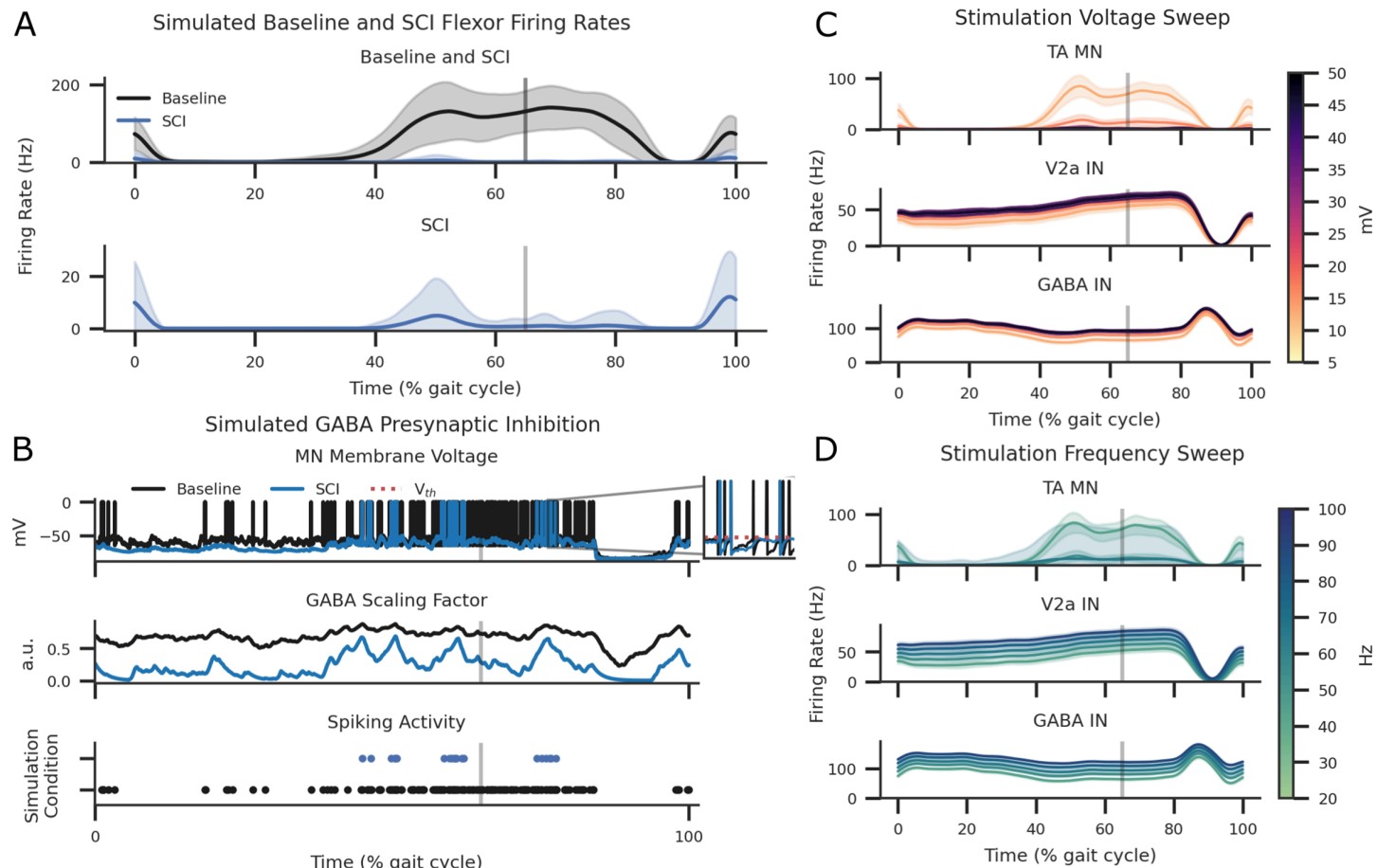

**Fig 4**. **The effect of increased GABA synapses and excessive stance induced inhibition on TA MNs in addition to the effect of ES stimulation voltage and frequency on the Baseline condition stepping across each neuron are shown. Shaded areas represent standard deviations across seeds.** (A) Average and standard deviation plots of eight steps during Baseline and simulated SCI conditions. The vertical grey line separates the stance (left of the grey line) and the swing (right of the grey line) phases, estimated at 65% of the gait cycle [85]. (B) SCI simulation of a single motoneuron receiving presynaptic inhibition by local concentration of GABA transmitters during a single step. Increased GABA scaling factor reduced the number of spike events near TA MN threshold. All shaded areas are standard deviations across seeds. (C) ES stimulation intensity sweep applied at 5, 10, 20, 30, 40, and 50 mV at a 40 Hz frequency. (D) ES stimulation frequency sweep applied at 20, 40, 60, 80, and 100 Hz at 10 mV amplitude. All ES was simulated with Poisson inputs to all flexor and extensor afferent axons.

GABA IN firing rates were increased while receiving ES inputs (S5 Fig and S5 Table). V2a IN firing rates were equal between Baseline and SCI since it did not receive GABA IN synapses (S6 Fig and S5 Table). Simulating SCI serotonergic agonist activity by reducing the leak conductance of V2a INs significantly increased the V2a firing rates (S6 Fig), though this only slightly increased firing rates in MNs by a factor of 1.4 (S5 Table). This facilitatory effect was abolished when combined with ES, see Figs 6, 5A, and 5C. Without BWS, 5-HT application significantly increased V2a interneuron firing rates ($p < 0.0001$; Figs S6 and 6) but did not restore TA motoneuron activity to Baseline levels (Fig 5A and 5C). The facilitatory effects of 5-HT on TA motoneurons and V2a interneurons were abolished when ES was applied, coinciding with increased GABA interneuron activity (S5 Fig and S5 Table). Combining 5-HT with ES in the absence of BWS further reduced motoneuron excitation below that observed in the SCI condition.

BWS locomotion with SCI increased overall flexor activity to averages greater than the Baseline condition ( Fig 5A, 5C, and 5D and Table 6). This was further amplified with the introduction of 5-HT. Applying ES without 5-HT smoothened

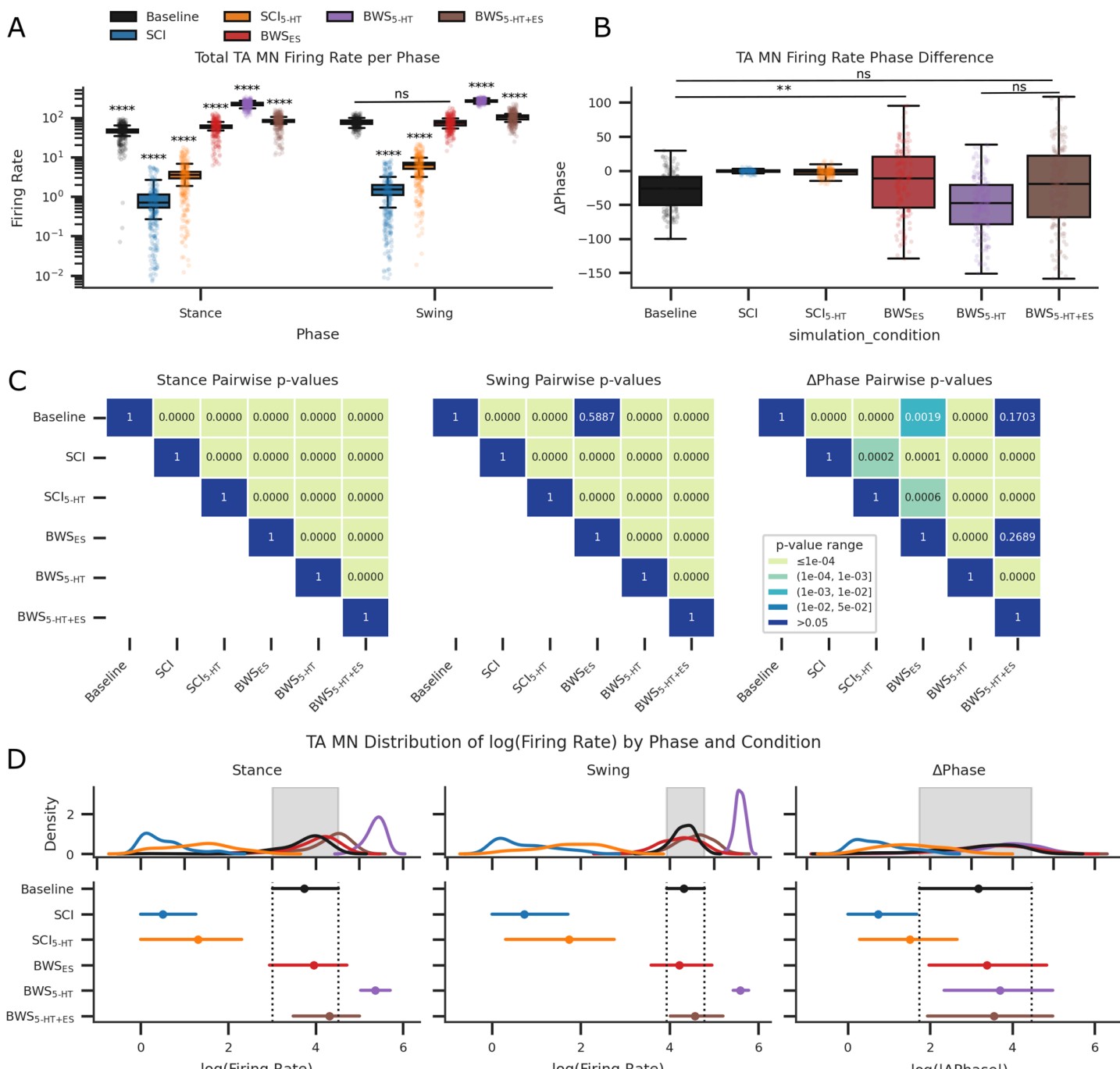

Fig 5. **Box-and-whisker plots with overlaid scatter data points for each step at each seed across conditions.** (A) Box-and-whisker plots were averaged over all seeds, yielding eight step datapoints and overlayed with scatter points for each phase. All swing and stance flexor activity was non-normal and significantly different with the exception of Baseline to BWS$_{ES}$ during swing phase after paired t-tests. (B) Plots the overlayed box-and-whisker plot and scatter points for the phase difference between stance and swing. No statistical difference was detected in the mean firing rate phase differences between SCI and SCI$_{5-HT}$. (C) Displays the heatmap p-values for pairwise t-tests. (D) Plots the kernel density estimate to visualise the distribution of log-transformed observations for phases and their difference. TOST testing on log-transformed data resulted in no meaningful difference between Baseline and BWS$_{ES}$ condition for both stance and swing phase. All TOST for phase difference calculations returned meaningfully large differences across all conditions.

**Table 6**. **Statistical calculations of seed-aggregated, phase-specific firing rates across experimental conditions for TA MN following z-score based outlier removal (values exceeding ±3 standard deviations).** Reported results include mean, standard deviation (SD), and kurtosis for each locomotor phase (stance and swing) and their difference.

| Phase | Simulation Condition | Mean | SD | Kurtosis |
|---|---|---|---|---|
| Stance | Baseline | 46.69 | 19.89 | -0.50 |
| | SCI | 0.91 | 1.24 | 4.73 |
| | SCI$_{5-HT}$ | 3.81 | 3.57 | 1.92 |
| | BWS$_{ES}$ | 59.08 | 26.91 | -0.41 |
| | BWS$_{5-HT}$ | 218.41 | 44.93 | -0.34 |
| | BWS$_{5-HT+ES}$ | 81.81 | 32.67 | -0.39 |
| Swing | Baseline | 76.96 | 19.75 | -0.14 |
| | SCI | 1.57 | 1.91 | 2.01 |
| | SCI$_{5-HT}$ | 6.26 | 4.81 | 0.48 |
| | BWS$_{ES}$ | 73.90 | 32.01 | -0.03 |
| | BWS$_{5-HT}$ | 267.45 | 30.13 | -0.31 |
| | BWS$_{5-HT+ES}$ | 103.13 | 38.64 | -0.14 |
| ΔPhase | Baseline | -29.88 | 26.49 | -0.68 |
| | SCI | -0.73 | 2.14 | 1.81 |
| | SCI$_{5-HT}$ | -2.65 | 5.98 | 0.74 |
| | BWS$_{ES}$ | -16.15 | 48.74 | -0.51 |
| | BWS$_{5-HT}$ | -50.84 | 42.49 | -0.47 |
| | BWS$_{5-HT+ES}$ | -22.85 | 59.04 | -0.50 |

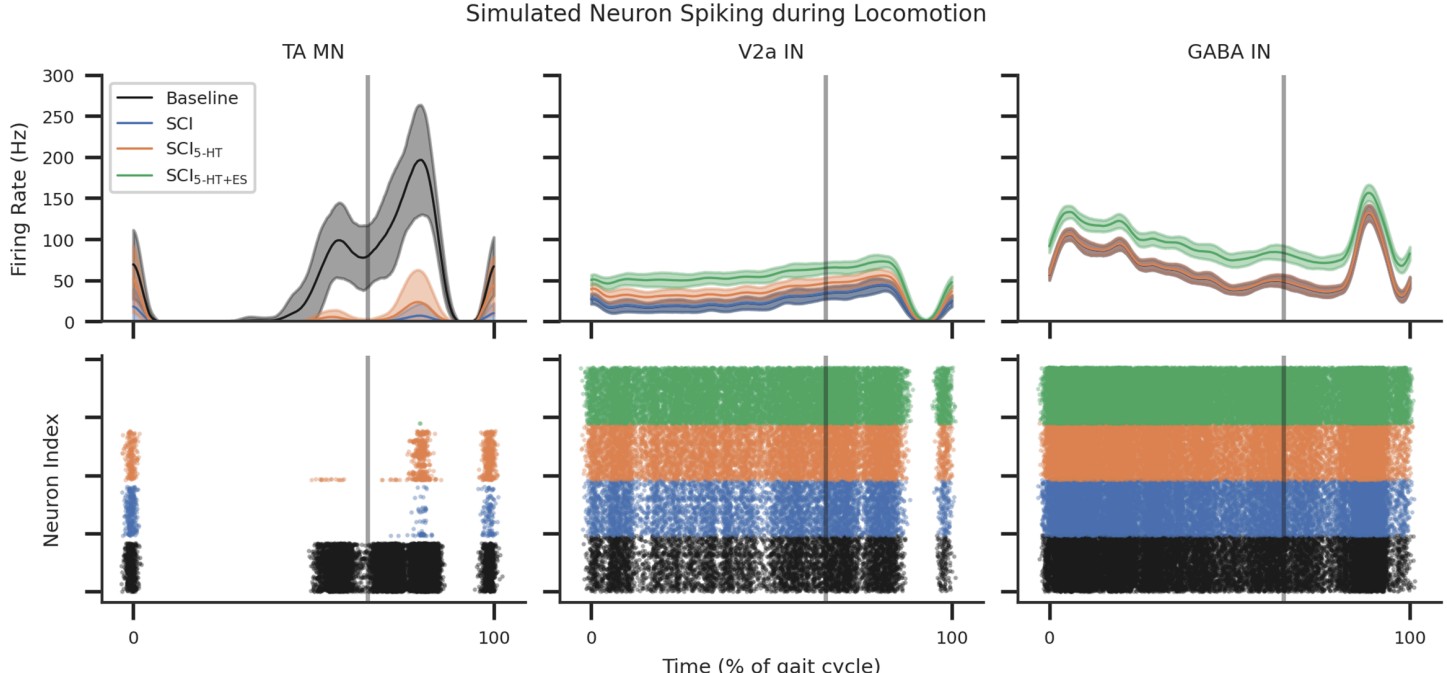

**Fig 6**. **MN, V2a IN, and GABA IN firing rates during an example step, averaged over 32 seeds and shaded SD.** The effect of SCI and SCI while receiving serotonergic agonists, (5-HT), and the combination of 5-HT and ES (5-HT+ES) were compared. The top row illustrates the population firing rates, and the bottom row shows the neuron population spike activity during the gait cycle.

the output of MN activity, returning MN activations to Baseline. Combining 5-HT and ES further increased peak activity during the swing phase and reduced activations during stance phases, see Figs 7, 5A, and 5C. By re-introducing ES the

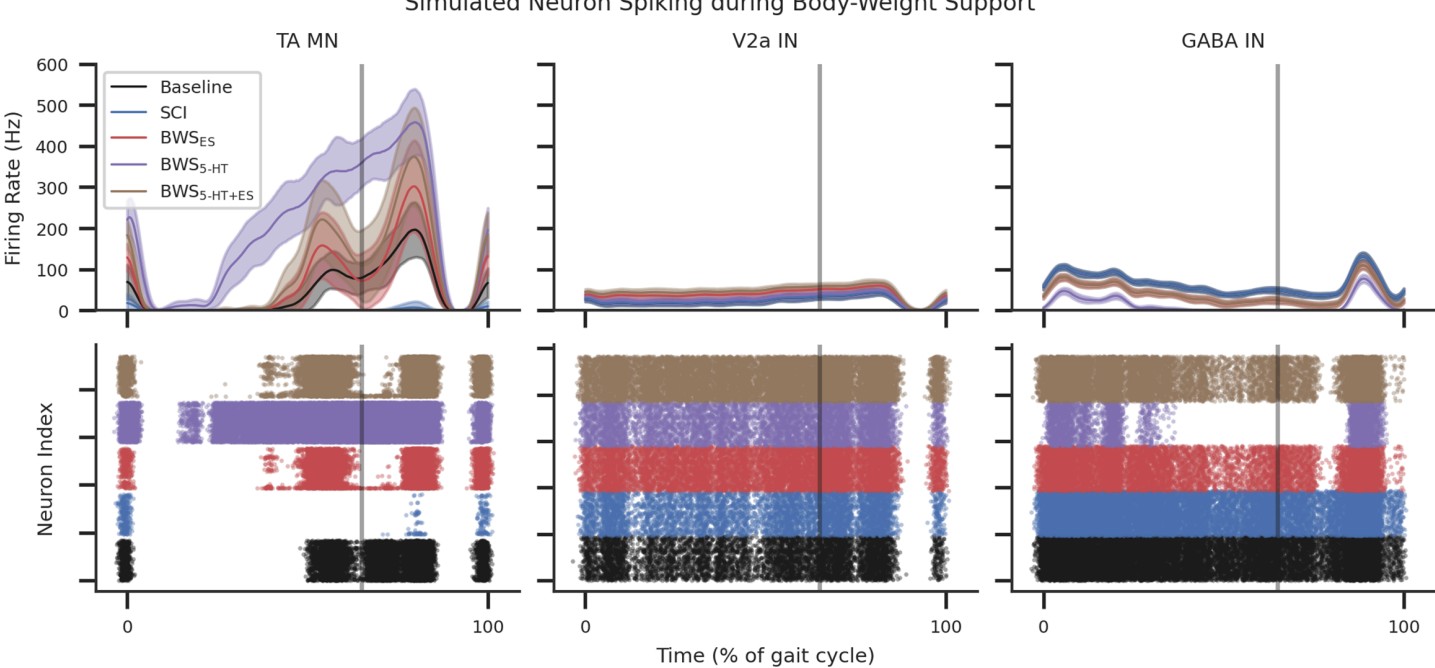

**Fig 7**. **MN, V2a IN, and GABA IN firing rates during an example step during simulated BWS locomotion averaged over 32 seeds and shaded SD.** The effect of BWS while receiving serotonergic agonists (5-HT) and the combination of 5-HT and ES (5-HT+ES) were compared. The top row illustrates the population firing rates and the bottom row shows the neuron population spike activity during the gait cycle.

$BWS_{5-HT}$ condition, the inter-step differences in firing rate area under the curve (AUC) became statistically insignificant between $BWS_{ES}$ and $BWS_{5-HT+ES}$ Fig 8D.

The reduction in flexor afferents during BWS decreased 5-HT modulated V2a IN activity towards SCI levels, though this effect did not reach statistical significance (S6 Fig and S5 Table). These changes were reversed under ES modulation. GABA IN activity was reduced in response to diminished stance-phase EMG during BWS Fig 7. Introducing ES restored GABA IN activity toward Baseline levels. GABA activity remained equivalent between $BWS_{ES}$ and $BWS_{5-HT+ES}$ as 5-HT did not provide additional modulation; refer to Figs 7 and S5.

To further quantify how these neuromodulatory conditions affected the temporal dynamics of locomotion, inter-step differences were analysed across the Baseline and $BWS_{ES}$ conditions by comparing firing rate, Fano factor (coefficient of variation), and AUC ( Fig 8B, 8C, 8D, and Table 7). Permutation testing showed the greatest similarity in firing rates during active bursting periods in both stance and swing phases, whereas the Fano factor similarity was highest during quiescent periods (Fig 8B). A relative t-test revealed that 62.5% of stepwise AUC values differed significantly between Baseline and $BWS_{ES}$, compared with only 25% of Fano factor measurements, see Fig 8C and Table 8.

No significant AUC differences were found between Baseline and $BWS_{ES}$ or between $BWS_{ES}$ and $BWS_{5-HT+ES}$. Despite comparable firing rate profiles, TA MN spike-train variability differed between Baseline and $BWS_{ES}$ during active spiking periods (see Figs 5C, 5D, and 8).

## Discussion

A biologically constrained SNN model of the flexor reflex circuit was developed to investigate the integrative mechanisms between sensory and neuromodulation inputs to the spinal cord. Analysis of the stance and swing phases across eight steps under SCI conditions revealed serotonergic agonists re-excited V2a INs and TA MNs after SCI. Nonspecific ES

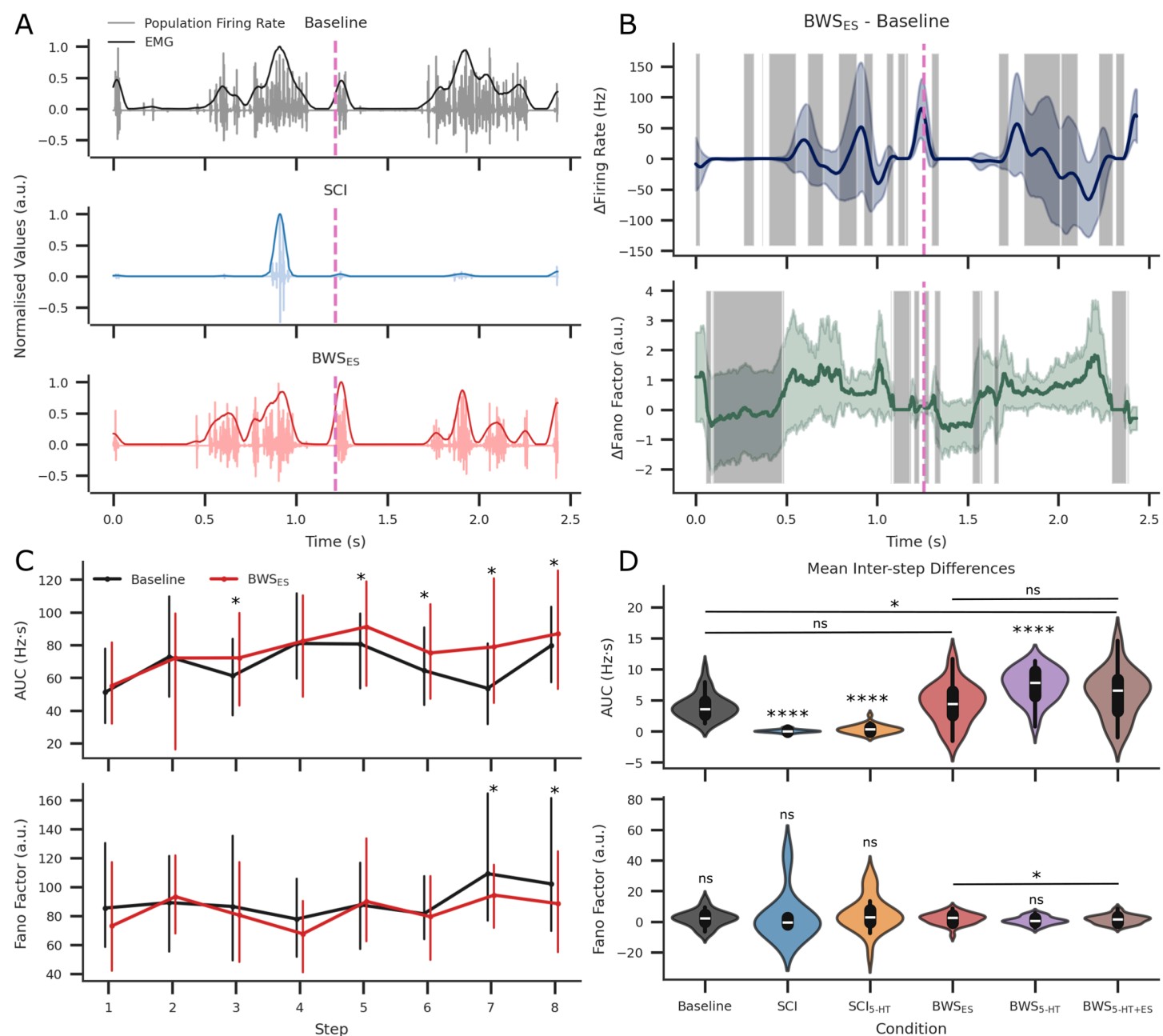

**Fig 8. Using the population firing rates across steps and seeds, Area Under Curve (AUC) and Fano factors were calculated after downsampling by a factor of 10.** (A) Population firing rates and EMG simulations for the Baseline, SCI, and BWS$_{ES}$ conditions. The pink vertical dashed line indicates the end of the first step and the beginning of the next. All y-axis values were normalised for easier viewing. (B) The firing rate and Fano factor difference between the Baseline and BWS$_{ES}$ conditions are plotted for the same step intervals as (A). Shaded in grey are areas that are not statistically significant after hierarchical permutation testing within each step (p > 0.05). (C) The step-by-step AUC and Fano factor with seed-aggregated means and highest density interval are plotted for the Baseline (black), SCI (blue), and BWS$_{ES}$ (red) conditions. (D) Inter-step differences in AUC and Fano factor were calculated for each seed and condition, providing a dependent sample size of 32. Conditional differences were pairwise t-tested and pairs with significance are shown. Baseline and BWS$_{ES}$ showed no significant difference in AUC. However, Baseline and BWS$_{ES}$ was significantly different when tested using the inter-step Fano factor.

**Table 7**. **Grand mean and standard deviation results of interstep differences aggregated over seeds for AUC and fano factor in the TA MN.** Some steps did not return any spikes and so were not included in calculation for fano-factor.

|  | Simulated Condition | Count | Mean | SD |
|---|---|---|---|---|
| AUC | Baseline | 32 | 4.08 | 2.04 |
|  | SCI | 32 | 0.07 | 0.26 |
|  | SCI$_{5-HT}$ | 32 | 0.34 | 0.71 |
|  | BWS$_{ES}$ | 32 | 4.53 | 3.21 |
|  | BWS$_{5-HT}$ | 32 | 7.47 | 2.61 |
|  | BWS$_{5-HT+ES}$ | 32 | 6.13 | 3.82 |
| Fano Factor | Baseline | 32 | 2.13 | 5.23 |
|  | SCI | 9 | 3.29 | 15.50 |
|  | SCI$_{5-HT}$ | 26 | 5.22 | 11.25 |
|  | BWS$_{ES}$ | 32 | 2.05 | 3.67 |
|  | BWS$_{5-HT}$ | 32 | 0.79 | 2.62 |
|  | BWS$_{5-HT+ES}$ | 32 | 1.55 | 2.92 |

**Table 8**. **Grand mean and standard deviation (SD) of Fano factor values during active bursting across seeds (n = 32).** Relative t-test showed significance between groups.

|  | Baseline | BWS$_{ES}$ | p-value |
|---|---|---|---|
| Fano factor | 0.66 ($\pm$ 0.06) | 1.46 ($\pm$ 0.22) | < 0.0001 |

application to proprioceptive afferent axons amplified the effects of reciprocal inhibition, further accentuating excitatory peaks and inhibitory valleys. Simulated BWS locomotion, implemented by adjusting the EMG gain in afferent equations, decreased presynaptic inhibition and consequently restored TA MN firing rates. Combining ES with BWS locomotion produced an activation profile more closely resembling Baseline activity across and within step phases.

Historically, ES has been applied for chronic pain management and spasticity reduction [75,136,137]. The activation pathways of spinal cord stimulation for pain management are understood to be via the large-diameter dorsal column and root fibres that carry propriosensory, mechanosensory, and nociceptive information [138]. GABA INs activate and depress afferent nociceptive signals by antidromic activation of the dorsal column at frequencies, electrode positions, and stimulation amplitudes similar to those of ES for sensorimotor recovery [139,140]. Similarly, ES applied for spasticity reduces the excitatory inputs to MNs through the proprioceptive inhibitory pathways [141,142]. Yet literature in SCI motor recovery places an intense focus on excitation [31,143,144].

Given the heightened inhibitory state of the injured spinal cord, it seems intuitive to return excitation to depressed neurons. However, results from this study suggest that activating the spinal cord with the same proposed mechanisms as pain and spasticity modulation equally activate inhibitory pathways, strengthening an already maladapted inhibition-dominant circuit [86,145]. A more refined and nuanced approach needs to be considered in order to return the required balance of excitation and inhibition to allow phasic activity to propagate. Results in this study suggest that appropriate sensory information must be provided to drive flexor network plastic adaptation towards a less inhibited and more task-specific tuned state. Tonically depressing or exciting, the spinal circuits do not provide the necessary sensory information to provide that plastic tuning [146,147]. This is in agreement with the requirement of propriosensory information for locomotor recovery after SCI [32–34].

The normal sensory processing occurring within the injured spinal cord becomes more stochastic and lacks the necessary bias required to perform the task [133]. As a result, pre-motor polysynaptic connections play a more active role in the expression of muscle tone and activity [148]. By establishing an appropriate balance in excitation and inhibition, repetitive sensory information can reinforce appropriate synaptic adaptations towards a more functional spinal state [70,149]. Results in this study investigate the synaptic mechanisms by which this can be accomplished, and provide a step towards understanding how BWS treadmill training works in concert with neuromodulation therapies [150–152].

The greater deviation in the normalised firing rate indicates increased variability in neuronal population activity after SCI (Fig 4A and 4B). This variability may arise from insufficient excitatory drive to maintain coordinated inhibition and excitation in phase with incoming sensory inputs [133]. Enhancing the excitability of motoneurons and premotor excitatory neurons, while reducing stance-phase inhibition through BWS, can therefore mitigate excessive inhibition of flexor pathways [153–156]. Once the network reaches a sufficiently excitable state that allows the timely propagation of phasic step information, subthreshold electrical stimulation may act as a stabilising mechanism - reinforcing relevant pathways without inducing excessive excitation or inhibition [74,120,121].

Interestingly, firing rate and Fano factor dynamics diverged during burst activity across steps. Although firing rate profiles were similar between steps, the coefficient of variation in spiking activity shifted toward $BWS_{ES}$ values (Fig 8B and Table 8). This divergence may reflect increased neuronal excitation resulting from reduced GABAergic presynaptic inhibition and enhanced facilitation through ES inputs and their downstream synapses. These findings suggest that sensory afferent input combined with BWS and ES produced a broader and more diverse activation of neuronal populations compared with Baseline conditions. The return of stochastic bursting has been linked to improved stepping recovery [133], yet the origins and functional implications of this variability during locomotor rehabilitation after SCI remain poorly understood.

Computational studies such as this are limited in their ability to generalise due to the estimates and tuning that are required to generate the model itself. The simulated flexor reflex loop's SNN architecture is simplistic compared to the complex bidirectional information exchange between the contralateral sides [157,158]. Although the cells were modelled from experimental data, there are errors and missing variables within experiments that have a carry-over effect on computational models. This study utilised LIF and AdEx equations to reduce computational burden and improve simulation runtime. Though previous efforts have incorporated the same approach [69,74], mathematical approximations of firing patterns limit the generalisability [159,160]. However, even with simple estimations, a computational model could provide new hypotheses about the inner workings of neurological systems and unlock novel recovery protocols [140,161].

Future studies could include the experimental verification of these findings via electrophysiological or genetic ablation experiments in rodent models under BWS neuromodulated locomotion contexts. Moreover, extending the model to enable investigation of previous electrophysiological results that uncovered correlations between the appearance of long-latency polysynaptic potentials and recovery of locomotion in spinal rats [35,65,119,121,134,162]. Re-emergence of uninterrupted late-response polysynaptic potentials may be the expression of increased excitability in local spinal networks, re-balancing the inhibitory dominant pre-motor circuit [133,148]. Functional recovery may be mediated by increased magnitude in polysynaptic activity, compensating for the loss in direct excitation [70]. Conversely, this model could be extended to investigate the effects of ES in returning inhibition in a hyperexcitable environment, such as that of spasticity [23,137]. Currently, there are a diverse number of stimulation paradigms for different causes of spasticity with several hypotheses for the mechanism of action [163]. Such hypotheses cover the inhibition of Ia afferents and stimulation of $A\beta$ fibres to diminish the effect of overactive spinal reflex, reducing muscle tone [164]. Alternative hypotheses suggest the activation of presynaptic inhibition and inhibitory networks within the dorsal column [165]. Finally, extending the computational model to include neuroplastic dynamics could uncover the relationship between neuromodulation and neuroplastic adaptations [118,166,167]. Investigating these effects within an extended computational model would be worthwhile.

## Conclusions

The development of a biologically constrained SNN has provided insights into the mechanistic basis of sensory and neuromodulatory integration after SCI. Simulations suggest that BWS locomotion in conjunction with ES returns phasic flexor coordination in an inhibition-dominant environment. In contrast, serotonergic agonists alone increased sensory-driven flexor activation but did not reestablish baseline excitation, while the combination of ES and 5-HT in the absence of

BWS produced network overexcitability. These results highlight that recovering TA MN activity after excessive GABAer-gic presynaptic inhibition depends on maintaining a dynamic balance between excitation and inhibition within spinal circuits.

Although the current model incorporates key physiological constraints, it remains limited by simplifying assumptions and an incomplete representation of spinal architecture. Future experimental validation will be essential to strengthen the reliability of these inferences. Overall, this work presents a computational framework for probing the combined effects of neuromodulation and sensory afferents on spinal network dynamics, supporting the design of targeted locomotor rehabilitation strategies.

## Supporting information

**S1 Fig. Visualisation of synapse connection rules.**
(TIFF)

**S1 Algorithm. Leaky Integrate-and-Fire (LIF) neuron model used to simulate membrane potential dynamics.**
(PDF)

**S1 Table. TA and GM afferent axon tuning performance measured by Pearson correlation coefficient (CC) and mean absolute error (MAE).** All correlations were significant ($p < 0.05$).
(PDF)

**S2 Algorithm. Synthesis of Rat EMG From Binary Motor-Unit Action Potential (MUAP)**
(PDF)

**S2 Fig. Simulated AdEx LIF TA MN single spike**. (A) and tonic burst (B) response after receiving a 20 ms and 200 ms stimulation pulse at 670 pA, respectively.
(TIFF)

**S2 Table. Bayesian Linear Mixture Model checks for convergence across all conditions.**
(PDF)

**S3 Table. A hierarchical Linear Mixed Model (LMM) was created for whole step extracted firing rate.** Equivalence was established across seeds if $P(|\sigma| < \epsilon) > 0.9$.
(PDF)

**S3 Fig. Simulated AdEx LIF V2a IN tonic spiking response after receiving a 200 ms stimulation pulse at 30 pA.**
(TIFF)

**S4 Fig. Pairwise Pearson correlation results across conditions after mean aggregation across seeds.** Steps were concatenated to a single array. All correlations were significant ($p < 0.0001$).
(TIFF)

**S4 Table. Results of post-hoc pairwise comparisons between simulated conditions using seed aggregated step firing rate samples in the TA MN during stance and swing phase and their difference.**
(PDF)

**S5 Fig. Box-and-whisker plots of GABA IN firing rate activity across eight steps for all 32 seeds during Baseline and simulated conditions.**
(TIFF)

**S5 Table. Descriptive statistics of phase-specific firing rates aggregated over seeds across experimental conditions for V2a and GABA interneurons (INs) following z-score based outlier removal (values exceeding $\pm 3$ standard deviations, were removed).** Reported values include the mean, standard deviation (SD), and kurtosis for each locomotor phase (stance and swing) under all experimental conditions.
(PDF)

**S6 Fig. Box-and-whisker plots of V2a IN firing rate across eight steps for all 32 seeds during Baseline and simulated conditions.**
(TIFF)

## Acknowledgments

We would like to thank V. Reggie Edgerton and Parag Gad (University of California, Los Angeles) for the invaluable scientific discussions which led to the conception of the experiment. We would also like to thank Bryce Vissel (St. Vincent's Hospital and University of New South Wales Sydney) for having initiated SCI research at University of Technology Sydney (UTS). Thank you to Howe Zhu (UTS, University of Sydney) for the feedback and support over the years. Finally, thank you to Matthew Gaston of the UTS eResearch team for providing support for and system administration of the Interactive High Performance Computing (iHPC) facility. The computing power and stability of the iHPC system was paramount to running the experiment. This work was supported by the Research Training Program (RTP) scheme from the Department of Education and Training, Australia.

## Author contributions

**Conceptualization:** Raymond Chia.

**Data curation:** Raymond Chia.

**Formal analysis:** Raymond Chia.

**Funding acquisition:** Raymond Chia, Chin-Teng Lin.

**Investigation:** Raymond Chia.

**Methodology:** Raymond Chia.

**Project administration:** Raymond Chia.

**Resources:** Chin-Teng Lin.

**Software:** Raymond Chia.

**Supervision:** Chin-Teng Lin.

**Validation:** Raymond Chia.

**Visualization:** Raymond Chia.

**Writing – original draft:** Raymond Chia.

**Writing – review & editing:** Raymond Chia, Chin-Teng Lin.

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
