## [Decision Letter · Decision Letter 0]

8 May 2025

PCOMPBIOL-D-25-00024

Biologically-Constrained Spiking Neural Network for Neuromodulation in Locomotor Recovery after Spinal Cord Injury

PLOS Computational Biology

Dear Dr. Chia,

Thank you for submitting your manuscript to PLOS Computational Biology. After careful consideration, we feel that it has merit but does not fully meet PLOS Computational Biology's publication criteria as it currently stands. Therefore, we invite you to submit a revised version of the manuscript that addresses the points raised during the review process.

Please submit your revised manuscript within 60 days Jul 08 2025 11:59PM. If you will need more time than this to complete your revisions, please reply to this message or contact the journal office at ploscompbiol@plos.org. Please include the following items when submitting your revised manuscript:

We look forward to receiving your revised manuscript.

Kind regards,

Simon M. Danner, Ph.D.

Guest Editor

PLOS Computational Biology

Mark Alber

Section Editor

PLOS Computational Biology

**Additional Editor Comments :**

The reviewers found the topic of your manuscript compelling and recognized potential value of your modeling approach for understanding neuromodulatory interventions after spinal cord injury. However, all three reviewers recommended major revision, citing significant concerns. The most pressing issues include: (1) insufficient model detail and missing figures that currently prevent reproducibility and hinder interpretation; (2) inadequate physiological justification for key modeling assumptions, particularly the representation of neuromodulatory effects and the inclusion of specific neuron types; and (3) overinterpretation of results without sufficient support, especially, but not exclusively, concerning mechanistic claims/conclusions on neuromodulation and release of inhibition.

To move forward, a comprehensive revision is necessary. This should include a full and transparent model description, justification for all assumptions and simplifications, inclusion of all referenced figures, and additional data or simulations to support your conclusions. Please ensure that all reviewer comments, both major and minor, are addressed point by point in your revised submission.

**Journal Requirements:**

At this stage, the following Authors/Authors require contributions: Raymond Chia, and Chin-Teng Lin. Please ensure that the full contributions of each author are acknowledged in the "Add/Edit/Remove Authors" section of our submission form.

3) Your manuscript is missing the following section heading: Abstract.  Please make sure that the section heading levels are clearly indicated in the manuscript text, and limit sub-sections to 3 heading levels. An outline of the required sections can be consulted in our submission guidelines here:

5) We have noticed that you have uploaded Supporting Information files, but you have not included a list of legends. Please add a full list of legends for your Supporting Information files after the references list. Please also cite and label the supplementary table and figures as “S1 Table”, "S1 Figure", S2 Figure" and so forth.

**Reviewers' comments:**

Reviewer's Responses to Questions

**Comments to the Authors:**

**Please note that one of the reviews is uploaded as an attachment.**

Reviewer #1: In this computational study, the authors seek to model the impact of presynaptic inhibition of proprioceptive afferent terminals onto motoneurons during locomotor activity in various conditions after spinal cord injury including with epidural stimulation, serotonergist agonism, and body weight support.

The topic is interesting and the model, while simple in its composition, includes physiologically detailed modelling of proprioceptive afferent activity.

Major issues:

-The inclusion of V2a interneurons is not well motivated in the manuscript

-The writing could be improved in the Results section. Results and approaches are often presented without much background or context. For example, why simulate photostimulation of GABA neurons? Those intimately familiar with the study of these neurons may figure it out but other readers will not understand that set of simulations.

-The figures can also be improved through the addition of many more simulation results. There are no displays of the simulation output in terms of "locomotor" activity. We are only shown the results of the analysis of that output. It is hard to gauge the physiological relevance of the simulation if we cannot see how the model generates locomotor output.

-There is a lot of evidence that spasticity may arise from hyperexcitable motoneurons. To gain a better understanding of how presynaptic inhibition could impact motor activity after spinal cord injury, it would be useful to model possible hyperexcitable motoneurons and see what would be required of presynaptic inhibition to counteract this potential maladaptive motoneuron change.

Minor issues:

-What does "target neuron reverse potential" mean? By convention, it is the synapse rather than the neuron that has a reversal potential

-Would it be possible to show a f-i curve for the GABA INs and V2a INs to show the consequences of using a AdEx model for those two cell types as opposed the F-i curve of IaINs or MNs? I guess it might be in figure 10, but it is never shown. Also, it is not clear what is the difference between the MN and V2a model. The firing looks tonic for both so not sure why require an AdEx LIF for GABA and V2as?

-In Fig 1F, the MN spiking is quite low-amplitude to be considered spikes, though the figure legend suggests that they are extracellular voltage activity. This is a bit confusing as why wouldn't intracellular activity be shown instead?

-Figure 2A, can you clarify what is being stimulated in the MEPs? It is all of the sensory afferents?

-It is not clear in which simulations the GABAergic connectivity to flexor MNs reduced the mean firing rate by 53.6 Hz. Is that in the lower panel of Figure 4? We'd suggest modifying the writing to better walk the reader through the simulations.

Reviewer #2: This study constructs a spiking neural network (SNN) model of the rodent tibialis anterior (TA), incorporating excitatory and inhibitory synaptic interactions, including GABAergic presynaptic inhibition. The neurons are represented by conductance-based, leaky integrate-and-fire (LIF) models. The authors claim to examine the modulatory effects of electrical stimulation (ES) and serotonin agonist (5-HT) in the context of spinal cord injury (SCI), both with and without body-weight-supported (BWS) locomotion, to investigate state-dependent changes in motor neuron excitability. The authors conclude that the synergistic interplay of BWS and ES suppresses excessive stance-induced GABAergic inhibition, shifting the system toward a stable dynamical regime where TA motor neuron firing rates approximate baseline homeostasis. The reviewer is not convinced that single-conductance LIF neuronal models can adequately represent all conditions mentioned, and finds the evidence supporting the article’s claims insufficient. Addressing the concerns outlined in the Major and Minor sections below may change this assessment, but would require a major revision with detailed justification for most of the model’s assumptions.

Major.

The model descriptions lacks sufficient detail, making it impossible to reproduce the results based on the provided description. Additionally, understanding the findings is challenging since one of figures (figure 9) is missing, which appears essential for understanding the results. Providing more detailed methodology and including these figures would greatly enhance clarity and help reproducibility.

The claim that the “biologically constrained spiking neural network (SNN) was developed” should be supported by the list of the parameters/constrains taken from the experimental measurements. The term 'biologically constrained' appears nine times in the manuscript, each followed by references. However, the manuscript lacks detailed descriptions of the specific parameters adopted or modifications made based on the referenced experimental data. I would also like to emphasize the importance of clearly distinguishing in the text between measured values and assumed or tuned model parameters from those references.

Formal Definition of “a baseline level of locomotive activity” versus an inhibitory-dominant and body-weight supported (BWS) SCI state is needed.

How is Electrical stimulation modeled? What is the justification?

“Ia and II TA and gastrocnemius medialis (GM) muscle afferent signals were calculated by using musculoskeletal and muscle spindle models during locomotion, described in previously validated computational models (62).” The details must be provided.

“The present study describes, for the first time, the release of GABAergic inhibition on flexor MNs as a potential mechanism underlying BWS treadmill training.” This claim must be clearly supported by the results. This reviewer failed to find the evidence.

“The results demonstrate the synaptic mechanisms by which neuromodulatory therapy tunes the excitation and inhibition of ankle flexor MNs during locomotion for smoother and more coordinated movement.” The modeling data should be described in sufficient detail for a reader to agree with this conclusion.

In the model descriptions the equations do not have subscriptions referring to a specific neuron in the circuit and this is impossible to follow. What is the meaning of the variable u in (5) and w in (6)? Is this the same variable?

Is capital Vth in tables correspond to low case v^th in equation 5? Where is Vrest and τrefractory used? Could full description of a LIF model be provided for each neuron type so the models could be reproduced? What is the meaning of the variable a?

“To simulate serotonin agonist response, MN gL was reduced by 40% during SCI experiments and 15% during BWS experiments (62, 85, 96). “ Could you justify that just one parameter is sufficient to represent these conditions?

“See Fig. 9 for MN responses with varying stimulation pulse widths.” Figure 9 is missing.

Page 7: “Refer to Fig. 1 A for an illustration of the entire network and figures 9 and 10 for tonic bursting neural firing.” The figures 9 and 10 are missing.

Page 7. “To simulate serotonin agonist response, MN gL was reduced by 40% during SCI experiments and 15% during BWS experiments.” The justification of these assumptions is needed.

Page 7. “See Fig. 9 for MN responses with varying stimulation pulse widths.” The figure 9 is missing.

Equation 7 is not clear. The parentheses which play no role suggest that the equation is not complete.

Equations describing synaptic currents including the parameters from the table 3 are not provided. Why the table 4 has a parameter values for p? Is this the variable p from the equation 7? Does this contradicts the figure 1 D?

In this manuscript, it is a serious problem that the variables representing different entities have the same name.

“Synaptic connections, with the exclusion of GABA, were determined by probabilities specified in table 4.” Were these synaptic connections constantly active?

“SCI condition GABA connections were increased by 160%...” Did you increase the conductances?

“GABA connections were tuned to match previous baseline responses (59, 62).” What are these responses? What does it mean? What parameters did you tune and how?

“Poisson distributed neural groups simulated subthreshold stimulation with user-specified input rates.” What is a neural group? What is a user-specified input rate? How are is stimulation modeled?

How is EMG signal generated?

Figure 2. How was stimulation intensities ranging from 0 – 600 µA encoded in the model?

“Results qualitatively match reported findings.” What do you mean by this?

Figure 2B. What is depicted on the right panel?

Page 6. “The ratio of extracellular potential during control and photostimulation conditions was within range of experimental results.” How do you measure the extracellular potential in LIF model?

“The burst activity during locomotion fell within the range reported in experimental data.” What is the measure used to make this claim?

Page 6. “Applying 5-HT+ES increased V2a IN and GABA IN activation, reducing the MN excitation to below SCI.” The authors should describe and justify precisely what are the changes in every parameter for each considered condition.

Minor

Importance of figure 1C is not clear.

Velocity in the equation 1 has the same variable name v as the membrane potential in the following equations. Assigning a different letter to it would be helpful.

Page 6. “The simulation time step was set to 5 µs and Euler approximations were used for ordinary differential equation solving. A total of 9 steps were simulated, where gait stance and swing phases were split at 65% of the gait cycle (67).” It would be easier to follow if the locomotion steps are distinguished from the time steps in the first sentence.

Page 3.

Cm 0.1 µF/cm2 why is so small? It is 10 time smaller than usually used in models.

Figure 1A The circle marking inhibitory input from GABA to motoneuron collocates with excitatory inputs from Flexor-Ia and V2a creating a confusing marker. Could you make these connections distinguished?

Figure 1 DE. Could you mark the voltage as a postsynaptic potential? Why -75 mV is chosen as a rest potential? This value does not refer to any neuron in the tables.

Figure 1F. Why time starts from 1 s?

Figure 1F “Left illustrates the MN extracellular voltage activity with spiking activity indicated with red triangles.” The figure appears illustrating intracellular membrane potential.

Figure 2C. Remove extra article “the”.

Page 6. “Simulating SCI serotonergic agonist activity by reducing the membrane conductance of V2a INs and MNs only slightly increased firing rates in MNs.” Why do you simulate “serotonergic agonist activity by reducing the membrane conductance”? Would not agonist activity increase the membrane conductance?

Reviewer #3: The review is uploaded as an attachment.

**Have the authors made all data and (if applicable) computational code underlying the findings in their manuscript fully available?**

Reviewer #1: Yes

Reviewer #2: None

Reviewer #3: Yes

PLOS authors have the option to publish the peer review history of their article (what does this mean?). If published, this will include your full peer review and any attached files.

Reviewer #1: No

Reviewer #2: No

Reviewer #3: **Yes:** Beck Strohmer

**Figure resubmission:**
---

## [Decision Letter · Decision Letter 1]

14 Dec 2025

PCOMPBIOL-D-25-00024R1

Biologically-Constrained Spiking Neural Network for Neuromodulation in Locomotor Recovery after Spinal Cord Injury

PLOS Computational Biology

Dear Dr. Chia,

Thank you for submitting your manuscript to PLOS Computational Biology. After careful consideration, we feel that it has merit but does not fully meet PLOS Computational Biology's publication criteria as it currently stands. Therefore, we invite you to submit a revised version of the manuscript that addresses the points raised during the review process.

We look forward to receiving your revised manuscript.

Kind regards,

Mark Alber, Ph.D.

Section Editor

PLOS Computational Biology

Mark Alber

Section Editor

PLOS Computational Biology

**Journal Requirements:**

At this stage, the following Authors/Authors require contributions: Raymond Chia, and Chin-Teng Lin. Please ensure that the full contributions of each author are acknowledged in the "Add/Edit/Remove Authors" section of our submission form.

**Reviewers' comments:**

Reviewer's Responses to Questions

**Comments to the Authors:**

Reviewer #1: The authors have done a lot of work to address the revisions requested. Most of my revisions have been satisfied but there are some issues that remain.

Fig. 1C. Not sure what the dots are and the solid lines represent. Are solid lines the fits?

Fig. 3B. Not sure what the different lines represent. It makes it hard to compare to the experimental data

The photostimulation is still not motivated well. Is there a citation that can be referenced when this is first introduced in the Methods?

"The TA MN population firing rates between Baseline and SCI conditions were significantly different across time

(Fig. 5)". Can you clarify whether you mean Fig. 5A? Makes it easier to see where the info is.

The Results text describing Figure 5 should be improved as it is not clear what figures relate to the conclusions made.

Reviewer #2: Most of my concerns have been addressed, but a few issues remain to be fixed.

Concerning the state definitions, the SCI condition is reasonably well-defined with a specific 1.6x increase in GABA IN→TA MN synapses, though the implementation method could be clearer. However, the BWS state lacks a critical quantitative parameter—the actual magnitude of the scalar reduction in Ia and II afferent firing rates—making it insufficiently defined for reproducibility without consulting the cited reference.

Table 1: "V Intracellular membrane potential", please, remove intracellular, it is part of the definition of the membrane potential.

Page 5:"AdEx equation was defined per equation (5)." There are two equations marked by (5).

I think it is more accurate to call periodic spiking during a square pulse of injected current 'tonic spiking' rather than 'tonic bursting.

Response: "This typing error has since been amended (equation 6, p. 5)." Equation 6 still has parentheses which play no role.

Reviewer #3: The authors have adequately responded to the initial review and updated the manuscript in a satisfactory manner.

**Have the authors made all data and (if applicable) computational code underlying the findings in their manuscript fully available?**

Reviewer #1: Yes

Reviewer #2: Yes

Reviewer #3: Yes

PLOS authors have the option to publish the peer review history of their article (what does this mean?). If published, this will include your full peer review and any attached files.

Reviewer #1: No

Reviewer #2: No

Reviewer #3: No

**Figure resubmission:**
---

## [Editor Report · Decision Letter 2]

22 Dec 2025

Dear Dr Chia,

We are pleased to inform you that your manuscript 'Biologically-Constrained Spiking Neural Network for Neuromodulation in Locomotor Recovery after Spinal Cord Injury' has been provisionally accepted for publication in PLOS Computational Biology.

Best regards,

Mark Alber, Ph.D.

Section Editor

PLOS Computational Biology

Mark Alber

Section Editor

PLOS Computational Biology

---

## [Editor Report · Acceptance letter]

PCOMPBIOL-D-25-00024R2

Biologically-Constrained Spiking Neural Network for Neuromodulation in Locomotor Recovery after Spinal Cord Injury

Dear Dr Chia,

I am pleased to inform you that your manuscript has been formally accepted for publication in PLOS Computational Biology. Your manuscript is now with our production department and you will be notified of the publication date in due course.

With kind regards,

Anita Estes
